# Cancer Extracellular Matrix Proteins Regulate Tumour Immunity

**DOI:** 10.3390/cancers12113331

**Published:** 2020-11-11

**Authors:** Alex Gordon-Weeks, Arseniy E. Yuzhalin

**Affiliations:** 1Nuffield Department of Surgical Sciences, University of Oxford, Room 6607, Level 6 John Radcliffe Hospital, Headington, Oxford OX3 9DU, UK; 2Department of Molecular and Cellular Oncology, University of Texas MD Anderson Cancer Center, Houston, TX 77030, USA; ayuzhalin@mdanderson.org

**Keywords:** cancer, extracellular matrix, immunity, immune, immune checkpoint, therapy, metastasis, collagen, fibronectin, microenvironment

## Abstract

**Simple Summary:**

The extracellular matrix (ECM) is the acellular part of a tissue, a meshwork of extracellular material produced and secreted by cells into the surrounding medium. The major function of the ECM is to provide structural and support to the surrounding cells. The ECM of malignant tumours is characterised as disorganized with increased production of some components compared to normal tissues. In this review we discuss how cancer ECM affects the tumour immunity and how we could exploit the ECM to boost the anti-tumour immune response.

**Abstract:**

The extracellular matrix (ECM) plays an increasingly recognised role in the development and progression of cancer. Whilst significant progress has been made in targeting aspects of the tumour microenvironment such as tumour immunity and angiogenesis, there are no therapies that address the cancer ECM. Importantly, immune function relies heavily on the structure, physics and composition of the ECM, indicating that cancer ECM and immunity are mechanistically inseparable. In this review we highlight mechanisms by which the ECM shapes tumour immunity, identifying potential therapeutic targets within the ECM. These data indicate that to fully realise the potential of cancer immunotherapy, the cancer ECM requires simultaneous consideration.

## 1. Introduction

Advances in our understanding of tumour immunity have led to the development of novel therapies with which to treat cancer. The introduction of Chimeric antigen receptor (CAR) T-cells [1] and immune checkpoint inhibitors are notable successes [2], yet most patients demonstrate resistance to these treatments [3]. Cancers with excessive extracellular matrix (ECM) deposition are particularly resistant, indicating that the ECM plays a fundamental role in regulating cancer immunity.

The simplest explanation for this is that ECM density provides a physical barrier, preventing interaction between immune effector and tumour cells. Although this may be contributory, mathematical modelling suggests that the mechanistic relationship between the ECM and tumour immunity is more complex [4]. The ECM was also reported to regulate immune cell motility [5], myeloid polarization [6], T-cell phenotype [7], immune cell metabolism [8] and survival [9]. Even prior to cancer cell dissemination, alteration of the ECM in distant organs is responsible for shaping an immune microenvironment conducive for metastasis [10]. To fully harness the potential of immunotherapy, treatments must be developed that address the functional contribution of the cancer ECM to tumour immunity.

## 2. Extracellular Matrix Organisation in Healthy Tissue

The ECM consists of over 300 different proteins [11], with further diversity brought about through splice variation, post-translational modification [12] and protein-protein crosslinking [13]. Two classes of protein predominate; fibrillar proteins and glycoproteins [14]. These proteins interact with each other to form large macromolecular structures. Their interactions and the relative proportion of different proteins within the matrix enable generation of numerous microenvironments with specific physical and biochemical properties. Re-modelling results from physical stresses put upon the ECM as well as the action of growth factors on stromal cells which regulate ECM production and breakdown. These processes enable the ECM to continually modify its structure and physics to meet specific functional requirements.

The ECM can be considered to be organized into peri-cellular (glycocalyx), basement membrane and interstitial matrices. This organization is brought about through the incorporation of different ECM proteins into each region, leading to specific physical and functional properties.

### 2.1. Pericellular Membrane (Glycocalyx)

The peri-cellular matrix consists primarily of glycoproteins, glycolipids, proteoglycans and glycopolymers, with immense diversity in the monosaccharide moieties of each providing diversity in glycocalyx function. The glycocalyx forms an interface between the cell surface and underlying fibrillar matrix and is difficult to study because of its complexity and small volume in comparison to other regions of the ECM. Nonetheless, specific functions have been ascribed to its particular constituents. Expression of the proteoglycan mucin-1 within the glycocalyx, for example, enables cell movement by reducing the force required to deform the plasma membrane [15]. Expression of the glycosaminoglycan hyaluronin within the glycocalyx modulates interaction with the cell membrane protein CD44 which modulates stemness in epithelial cells [16] and interaction with numerous lymphocytic subsets [17]. Whilst a full description of the compositional and functional specification of the peri-cellular membrane is outside the scope of this review, we point the reader to several recent reviews on the topic [18,19].

### 2.2. Basement Membrane

One of the most consistent macromolecular ECM structures is the basement membrane (BM). The BM consists of type IV collagen and laminin networks linked by the glycoproteins nidogen and perlecan [20]. Laminin polymerisation is the first step in BM assembly, followed by cross-linking of laminin to type IV collagen. The laminins are a large family of glycoproteins consisting of at least 16 isoforms. Each laminin consists of three chains (α, β and γ), with 5 α, 4 β and 3 γ subtypes, enabling immense diversity in function. Laminin molecule expression in the BM is tissue-specific; endothelial BMs are composed predominantly of laminins α4β1γ1 (411) and 511, whereas colonic epithelial BM consists of laminin 332 and that of the glomerulus express laminin 521 [21]. Following laminin deposition, collagen IV molecules polymerise to form a second network that provides mechanical strength and stability [22].

The BM surrounds the basal aspect of epithelial and endothelial cells in virtually every organ, providing a point for cellular adhesion, modulating stemness, partitioning epithelial and stromal compartments and regulating epithelial-stromal cell communication. Measurements of the physical properties of the BM are technically challenging but appear to vary for BMs in different tissue type [23]. This is depen dent upon BM composition, the physical properties of the underlying stroma and interaction from epithelial cells. Changes in BM physics modulate epithelial cell phenotype through mechanotransduction pathways and the BM therefore plays an important role in the regulation of tissue homeostasis.

### 2.3. Interstitial ECM

The interstitial matrix, deep to the basement membrane, consists of fibroblasts, resident immune cells, vasculature and lymphatics embedded within a loose ECM of collagens I and III, elastin fibers, and glycoproteins [24]. Less abundant fibrillar collagens including types V and XI promote the nucleation of collagen fibrils [25]. These collagens maintain N-terminus pro-peptides which protrude between major collagen fibrils, limiting fibril width and potentially providing a site for interaction with other ECM molecules [26]. The fibrillar matrix scaffold resists tensile forces and provides a framework upon which stromal cells can move [27]. Stromal glycoproteins including versican, fibronectin, biglycan, lumican and perlecan, bind and modulate the function of growth factors and chemokines, buffer hydrogen ions and bind water molecules. Through these processes, they form a hydrogel that further modulates cell movement and provides tuneable resistance to compression.

Fibroblasts within the interstitial matrix exert tensional force on the fibrillar matrix, altering collagen fibril alignment and modulating the mechanical properties of the ECM [28]. Cross-linking of collagen fibers, catalysed by lysyl oxidase (LOX) enzymes provide further regulation of stromal physics [29]. The balance between matrix degradation and production is regulated by the competing actions of matrix metalloproteinases (MMPs) and tissues inhibitors of MMPs (TIMPs), as well as growth factors that modulate fibroblast activity, including TGFβ [30], fibroblast growth factors [31] and NOTCH ligands [32].

## 3. Extracellular Matrix Organisation in Cancer

The development and progression of cancer is associated with significant changes in the composition, structure and physical properties of the ECM within both the stroma and BM. Transcriptomic and proteomic comparisons of cancers and their normal tissue counterparts demonstrate differences in the abundance of both fibrillar collagens and glycoproteins [33,34,35]. Although most of the structural proteins within the cancer ECM are deposited by stromal cells, cancer cells contribute some ECM glycoproteins, ECM regulatory proteins and secreted factors [36,37,38,39,40]. Intriguingly, the ECM proteins deposited by cancer cells appear to play a significant role in disease biology, as it is the expression of these proteins that is best aligned with clinical outcome [36,40,41].

### 3.1. Peri-Cellular Membrane

Alterations in the composition of the peri-cellular membrane are frequently reported in cancer. The glycocalyx of cancer cells demonstrates enhanced glycosylation altering the physical properties of glycoproteins within the peri-cellular membrane. Increased expression of bulky glycoproteins within the glycocalyx of cancer cells serves to regulate the topography of cell surface receptors. This has been demonstrated for the glycoprotein Muc-1, whose expression concentrates integrin receptors within kinetic traps on the cell surface, promoting their interaction with ECM molecules in the basement membrane or interstitial matrix; a feature required for cell survival [42]. Other alterations in the cancer cell glycocalyx include increased deposition of hyaluronin, various heparin sulphate proteoglycans and members of the syndecan family of proteins [43]. The functional consequences of expression of these molecules specifically within the cancer glycocalyx remains elusive.

### 3.2. Basement Membrane

The BM undergoes significant alteration during cancer development and progression [23]. Cancer cells invade the BM by producing ECM remodelling enzymes, by making use of natural pores in the BM [44], or by forcing their way through it [23]. Invasion is also dependent upon contact that cancer cells make with type I collagen fibers in the underlying stroma [45], indicating that changes in the BM and stroma are required for disease progression.

Whilst being denuded in certain regions, the BM is excessively thickened in other areas where it is composed of sheet-like arrangements of laminins, fibronectin and collagens I, III and IV [38,46,47,48]. This structural arrangement effectively partitions tumours into cancer cells nests and stromal regions, whilst the alterations in BM physics associated with these structural changes promotes the malignant behaviour of cancer cells [49]. This partitioning may be particularly important for the regulation of tumour immunity as is discussed subsequently.

### 3.3. Interstitial ECM

Cancers display elevated fibrillar collagen deposition throughout the interstitial matrix [50,51,52]. From a structural perspective, collagen fibers are thicker, more organised and more highly packed in cancers than in healthy tissues [53,54,55] and in particular, the tumour regions in proximity to the BM display increased fiber density [56]. As cancers progress, stromal collagen fibers become increasingly aligned [55,57], particularly at the cancers edged; a process that promotes cancer cell invasion [58,59]. Elevated LOX expression drives excessive cross-linking of collagen and is associated with poor prognosis in a range of cancers [60], whilst LOX inhibition delays cancer progression in pre-clinical models of primary [33] and metastatic cancer [34], although to date, has failed to demonstrate efficacy in humans [35].

Changes in collagen fiber deposition and cross-linking alter the physical properties of the ECM, increasing stiffness and contributing to solid stress within the tumour [61]. Stiffness is a material property defined as the degree to which a material resists deformation in response to an applied force. Deposition of highly cross-linked, thickened, fibrillar ECM proteins results in the stiff matrix typical of cancer. Solid stress (force/area) results from a combination of rapid cellular proliferation within a cancer, cell-ECM interactions and massive ECM deposition, all of which interact with resistive forces from the surrounding normal tissue. These physical features are sensed by cancer and stromal cells through mechanotransduction pathways and inhibition of mechanotransduction pathways including FAK, Rock/Rho and YAP/TAZ have demonstrated efficacy in pre-clinical models [62]. In addition to sending biophysical signals to the tumor, collagen fibers may directly regulate the proliferation of cancer cells through binding the Discoidin Domain Receptor family (DDR1 and DDR2) expressed on the latter [63].

Expression of various glycoproteins within the ECM have also been linked to cancer progression. Tenascin C (TNC) is a glycoprotein involved in regulating embryogenesis, with expression in the adult limited to stem-cell niches [64]. Expression has been demonstrated in a range of cancers [65] and in breast cancer, stroma and cancer cell-derived TNC promotes metastasis by supporting a stem-cell phenotype in the metastatic niche [66]. Versican, an ECM glycoprotein that mediates tissue inflammation [67], is also over-expressed in cancer. In murine models, versican promotes the growth of lung cancer metastases through macrophage activation [68]. Finally, the glycoprotein fibronectin plays a prominent role in cancer biology and is expressed within the pre-metastatic niche, where it promotes the accumulation of bone marrow-derived cells that facilitate metastasis [69].

## 4. ECM Regulation of Immune Cell Migration

### 4.1. Structural ECM Proteins

In cancers of the colon [48], pancreas [56], breast [70] and lung [71], immune cells demonstrate heterogenous distribution, being found predominantly in the tumour stroma, rather than adjacent to, or amongst cancer cell nests. This correlates with a heterogenous structure of the collagen fiber arrangement throughout tumours which demonstrate loose packing in the stroma compared with regions around cancer cell nests where the fibers are thicker and closely packed [72]. This has led to speculation that the thickened ECM surrounding cancer cells limits immune cell motility, preventing immune-cancer cell interaction and thereby cytotoxicity [73]. This is supported by spatial analysis of cancers demonstrating correlation between loose ECM structures or the presence of specific ECM proteins and immune cell abundance [71,74,75,76,77], as well as the resistance of cancers demonstrating excessive ECM deposition to immune checkpoint inhibition [78,79].

Evidence for the importance of the ECM in regulating immune cell motility in the healthy state comes from in-vitro and in-vivo observations [80]. In 3D culture systems, both T-lymphocytes [81] and natural killer cells [82] move along fibrillar collagen networks in a process independent of integrin or protease activity. Their movement is amoeboid in nature with matrix porosity and nuclear deformability the primary determinants of migration [83]. Importantly, these experiments were performed in the absence of chemokine gradients indicating that the presence of collagen fibers alone is enough to guide immune cell movement.

In support of this, in-vitro migration of cytotoxic T-lymphocytes is reduced in concentrated collagen gels [84] or in regions of 3D collagen matrix where the fibers lack alignment [85] indicating that loose, well-aligned collagen fibers provide the optimal medium for T-cell migration. Further examples of the role played by the ECM in regulating immune cell motility are demonstrated in the lymphatic system, where T- and B-lymphocytes use amoeboid movement to migrate freely along the loose, reticular ECM fibers found in lymph nodes [86]. These observations indicate that the structural properties of the ECM provide important migratory cues for immune cells.

Live-cell imaging observations of the tumour microenvironment (TME) provide further support for these findings. In implantable (subcutaneous) thymoma tumours, activated T-cells migrate along collagen fibers identified through second harmonic generation imaging [87]. T-lymphocytes also utilise collagen fibers for motility within implantable lung tumours [71]. In both of the aforementioned settings, T-cell activation and recruitment to the tumour was artificially induced through implantation and so it is unclear whether similar T-cell migratory patterns would occur in a malignancy that develops spontaneously. Furthermore, although T-cells were seen to crawl along ECM fibers, their structural properties or contribution to ECM physics were not studied.

Almost all types of immune cells express Discoidin domain receptors (DDRs), a class of receptor highly responsive to collagen and key mediators of cell migration [88,89,90]. Through a series of experiments in was shown that blockade of DDR1 by a recombinant soluble protein DDR1:Fc leads to a reduction of T cell migration in 3D collagen in a dose-dependent manner [89]. Similarly, masking the binding sites on collagen with recombinant DDR resulted in a significantly reduced neutrophil migration through a collagen I–coated Transwell [91]. Furthermore, DDR1a-overexpressing THP-1 monocyte-like cells displayed an enhanced migration through 3D collagen matrix in comparison with mock-transfected cells [90]. Even though these findings clearly identify DDRs as key modulators of immune cell motility in the ECM, this topic remains entirely unstudied in relation to the cancer matrix. Considering that DDRs are therapeutically targetable receptor tyrosine kinases, their future investigation in the aspect of immune cell migration seems to be warranted.

Alterations in the ECM at future metastatic sites also governs immune cell motility. In KRAS mutant mice that develop spontaneous pancreatic cancer, the liver pre-metastatic niche displays enhanced deposition of type I collagen and fibronectin in association with F4/80^+^ macrophage and neutrophil infiltration [92]. This process was required for efficient liver metastasis formation and STAT3 activation was a requirement for ECM protein deposition in the pre-metastatic liver. STAT3 inhibition prevented the hepatic immune infiltrate and limited metastatic dissemination [92], indicating that liver fibrosis plays a key role in metastatic dissemination through alteration of local hepatic immunity.

ECM structure therefore regulates the movement and infiltration of immune cells into cancers (Figure 1) and future metastatic sites and emerging lines of evidence indicate that the abnormal ECM structure in cancers contributes to defective tumour immunity. T-cells cultured on freshly-isolated human lung and colon cancer explants for example, preferentially position themselves within the tumour stroma, rather than within cancer cell nests [48]. Motility of T-cells is greater in the stroma, where collagen fiber density is low, whilst collagenase treatment of tumour explants enabled T-lymphocytes to migrate into tumour cell nests [48]. In pancreatic cancer, collagen topography rather than chemokine concentration determined migration of tumour-activated T-cells, such that even in the presence of high levels of the T-cell chemoattractants CXCL10 or CCL4, motility was prevented by the presence of the high-density collagen fibers surrounding tumour cell nests [56]. In breast cancers, tumour regions demonstrating the highest level of collagen cross-linking in association with excessive lysyl oxidase activity and tissue stiffness demonstrate abundant macrophage infiltration [93]. Therapeutic ablation of macrophages in this setting inhibited metastatic dissemination, indicating that collagen crosslinking serves to drive metastasis through macrophage recruitment [93]. These contrasting data for T-cells versus macrophages indicate that different immune cell types may have unique abilities to navigate specific ECM structures. If this were the case, production of specific ECMs would give cancers the opportunity to exclude some immune cell subsets whilst enabling active infiltration of others. Therapeutic targeting of specific types of ECM rather than the tumour stroma more broadly, would therefore be beneficial.

The cancer ECM continuously undergoes a process of remodelling which in part involves proteolysis. Because the proteolytic fragments of various structural ECM proteins modulate immune cell function [94], proteolysis may serve as a mean of generating ECM protein fragments capable of regulating the movement of immune cells within the TME. In support of this theory, modulation of MMP and TIMP activity is associated with altered tumour immunity [95], whilst proteolytic fragments of type I collagen were responsible for M2 macrophage recruitment to the involuting breast; a high-risk microenvironment for breast cancer development [96]. Treatment of nude mice bearing subcutaneously injected breast cancer cells, with an adenovirus carrying the MMP-9 gene resulted in a significant reduction in tumour growth and an increase in intra-tumoural endostatin; a proteolytic fragment of collagen type XVIII [97]. Adenoviral-treated tumours displayed a florid innate immune response consisting of neutrophils and macrophages and the anti-cancer effect of MMP-9 delivery was abolished by therapeutic ablation of neutrophils [97], indicating that the immune influx dependent upon MMP-9 activity is responsible for delaying tumour progression. This function of collagen fragments is supported by analysis of subcutaneous lung cancer models which demonstrate a reduced influx of immunosuppressive myeloid-derived suppressor and M2 macrophage populations upon treatment with endostatin alongside tumour-primed T-lymphocytes [98]. Importantly, endostatin also inhibits angiogenesis and so it is likely that its anti-cancer effects result from both immune modulation and a reduction in tumour vasculature [99].

Neutrophils are also dependent upon the structural and physical properties of the ECM for motility. Neutrophils display a biphasic pattern of mobility when plated on polyacrylamide gels supplemented with fibronectin [100]. Culture in soft gels leads to reduced motility as a result of failure to generate traction forces due to reduced adherence, whilst motility in stiff gels is reduced because of increased adhesive forces. This indicates that maintenance of a moderate degree of stiffness enables optimum motility [100]. Currently a role for structural ECM proteins in the regulation of neutrophil motility in cancer has not been described, however, given the importance of neutrophils in cancer progression [101], this topic certainly deserves further scrutiny. Finally, binding of anti-collagen antibodies to collagen fibrils was localized to GPO/GPP region, which serves as interaction sequence for various immunologically associated molecules [102]. This could suggest an active involvement of collagen fibrils in the process of immune activation.

### 4.2. Non-structural ECM Proteins

It is not just the structural proteins within the ECM that regulate immune cell movement. Glycoproteins, proteoglycans and ECM regulating proteins also govern this process and because cancer cells are responsible for production of a number of these proteins, this gives them the ability to make unique changes to the way in which the cancer ECM regulates immune cell influx. In skin cancer models, the ECM glucosaminoglycan Hyaluronan and Proteoglycan Link Protein 1 (HAPLN1) has opposing effects on the motility of cancer and immune cells [103]. Aged fibroblasts lacking HAPLN1 expression generate a highly-aligned, stiffer matrix than that produced by fibroblasts from young patients that express HAPLN1, whilst in-vivo, addition of HAPLN1 to old mice reverses ECM fiber alignment in skin specimens. Treatment of melanomas with recombinant HAPLN1 inhibits tumour growth as a result of increased cytotoxic T-cell infiltration, leading to the conclusion that loss of HAPLN1 inhibits T-cell based rejection of melanoma through effects on ECM structure [103].

Tenascin-C (TNC), an ECM glycoprotein associated with poor prognosis cancer [65], also plays a role in the regulation of T-cell infiltration in cancer. T-cells failed to demonstrate amoeboid migration across ECMs derived from glioma cells, whereas movement was possible across ECMs derived from breast or hepatocellular cancer cell lines [75]. Short hairpin RNA knockdown of TNC in glioma cells enabled T-cells to move freely across their ECM indicating that TNC production by tumour cells is a means of paralysing infiltrating T-lymphocytes. These findings are supported by analysis of human gliomas which demonstrate accumulation of T-cells in the boundary between tumour and normal brain in association with TNC deposition [75].

Thrombospondin 1 (TSP1) is another ECM glycoprotein of relevance to immune modulation in cancer. In melanoma models, forced over-expression of TSP1 led to an influx of macrophages and a significant inhibition in tumour growth [100]. Macrophages within TSP1-expressing tumours were of an M1 (anti-tumour) phenotype and displayed elevated superoxide expression [104], implicating TSP1 in both the recruitment and polarisation of macrophages that reach the TME. Other ECM proteins promote the influx of immunosuppressive (M2) macrophages indicating selectivity of the ECM for specific myeloid cell subtypes. A secreted ECM glycoprotein collagen triple helix repeat containing 1 (CTHRC1) correlated with high density of M2-like tumour-associated macrophages (CD68^+^/CD163^+^) in patients with endometrial cancer [105], whilst CTHRC1 stimulated the recruitment of macrophages in vitro through integrinβ3/PI3K/Akt/CX3CR1 signalling [105].

Fibronectin is another ECM glycoprotein of relevance for tumour immunity. Fibronectin expression in the pre-metastatic niche promotes the recruitment of bone-marrow derived, haematopoietic, Vascular endothelial growth factor-1 (VEGFR1^+^) cells that ultimately promote metastatic dissemination through the production of factors that support the outgrowth of micrometastatic deposits [71]. The mechanisms through which the VEGFR1^+^ cells respond to fibronectin within the pre-metastatic niche remain unclear and much of this work is based on observation rather than mechanistic evaluation, but it nonetheless, it highlights the potential relevance of targeting ECM glycoproteins in order to affect cancer immunity.

The glycoprotein osteonectin, also known as secreted protein acidic and rich in cysteine (SPARC) is a protein that promotes ECM production and in normal health, regulates calcification of fibrillar collagen within bone. However, in melanoma, SPARC inhibition through stable transfection of short hairpin RNA targeting the SPARC transcript increased neutrophil recruitment, inhibiting tumour growth [106]. In transwell assays, melanoma-derived SPARC inhibited the migration of neutrophils but not lymphocytes or monocytes [106], suggesting that elevated SPARC production may allow melanoma cells to escape immune-mediated destruction by selectively inhibiting neutrophil influx. Finally, tumour-associated (M2) macrophages which promote a cancer stem cell phenotype and disease progression, are recruited to murine glioblastoma through expression of the ECM glycoprotein periostin (POSTN) [107]. Here POSTN promotes extravasation of TAMs into the tumour stroma through integrin-dependent mechanisms, rather than as a result of changes in ECM architecture [107].

Just as is the case for the structural ECM proteins, the proteolytic fragments of ECM glycoproteins can also serve as chemotactic cues for infiltrating immune cells. For example, colorectal cancers displaying proteolysis of versican demonstrate enhanced T-lymphocyte infiltration, whilst fragments derived from versican proteolysis (matriknes) promote dendritic cell accumulation, indicating a role for matrikines in the modulation of cancer immunity [108]. These findings, however, are contrasted by those in B16F1 murine melanomas, where inhibition of the proteolytic enzyme a disintegrin and metalloprotease with thrombospondin motif 1 (ADAMTS1), which is responsible for versican proteolysis, led to a reduction in tumour growth in association with increased intratumoural lymphoid and myeloid populations [109,110]. The cause of these disparate findings are unclear, but it should be borne in mind that ADAMTS1 has a number of targets within the ECM, so it is likely that its inhibition is not versican-specific and may have more widespread effects on ECM structure and function.

Fascinatingly, ECM proteins can also arrest the movement of immune cells by serving as binding substrates to inflammatory cytokines. Fibronectin serves this role through its ability to bind TNFα molecules. Whilst TNFα has no adhesion- or migration-promoting effects of its own, in-vitro findings demonstrate that when bound to fibronectin, it is able to halt CXCL12-driven migration of T-cells along fibronectin molecules [111]. In support of this, TNFα inhibition led to an influx of cytotoxic T-lymphocytes in experimental melanomas [112]. Fibrin, a protein involved in clot formation but deposited within the cancer ECM [113], also serves to arrest immune cells in the promotion of cancer progression. From a mechanistic perspective, soluble fibrin promotes adhesion of melanoma cells to endothelia via ICAM expression on the later, whilst under flow conditions, this interaction is strengthened fibrin-driven neutrophil recruitment and adhesion [114,115].

Finally, the galectins are a family of proteins that are of recognised importance for the regulation of tumour immunity [116]. When secreted into the ECM, galectins facilitate cell surface-ECM interaction and so their deposition within the ECM shapes various aspects of cellular behaviour. Early studies of the function of galectins demonstrated that these proteins inhibit adhesion of cancer and immune cells to structural ECM proteins including collagens, laminins and fibronectin [117,118]. These findings have been recapitulated in the TME also. In head and neck cancers, galectin-1 expression correlates with poor response to immune checkpoint inhibitors and in murine models, galectin-1 over-expressing cell lines prevented T-cell migration into the cancer [119]. Mechanistically, galectin-1 prevented T-cell infiltration into the tumour through up-regulation of immune checkpoint molecules on the tumour endothelium [119]. Galectin-3 also plays a role in immune cell migration within the tumour stroma. Secretion of galectin-3 into the ECM by tumour cells bound the inflammatory cytokine CXCL9, effectively negating the development of chemokine gradients that would otherwise guide T-cells towards the tumour cells [120]. Treatment of tumour-bearing mice with galectin antagonists normalised CXCL9 gradients, improving influx of CD8^+^ T-lymphocytes specific for tumour antigen [120].

## 5. ECM Regulation of Immune Cell Function

As well as modulating immune cell movement, the ECM also serves to affect immune cell function (Figure 2). For myeloid cell types this primarily involves effects on cell polarisation, whereas for lymphoid cells the ECM can have direct effects on activation state and cell cytotoxicity.

### 5.1. Myeloid Cells

Polarisation of myeloid cells, both macrophages and neutrophils is an important mechanism through which the TME modulates anti-cancer immunity [121,122]. Increasing evidence indicates that ECM physics, structure and protein composition mediate myeloid cell polarisation, however, the relative effects of each of these parameters appears context dependent and their relevance to cancer immunity is only just becoming realised. In-vitro, human macrophages demonstrate plasticity in response to alterations in matrix stiffness and GAG sulphation [123]. Post-translational sulphation of GAGs modulates their interaction with other ECM and cell-surface molecules, vastly increasing their functional repertoire [124]. Over a period of 6 days, when cultured on stiff matrices, macrophages exhibit a wound-healing (M2) phenotype, typified by increased expression of IL10 and a reduction in TNFα expression; effects that are reversed by GAG sulphation [123,125]. Contrasting results are identified in macrophages cultured in the presence of the inflammatory cytokines CCL2, IL6 and INFγ which demonstrate an impaired M1 phenotype following culture on artificial matrices composed of sulphated hyaluronic acid and type I collagen [126]. Macrophages cultured on highly cross-linked, stiff fibronectin-containing matrices display enhanced TNFα expression indicative of M1 polarisation [127]. An alternative mechanism through which ECM physics affects macrophage polarisation is via the intermediary action of mesenchymal stem cells, which promote M2 polarisation of macrophages when grown in stiff polyacrylamide gels through the secretion of soluble factors [128].

Lineage specification of monocyte/macrophage cell precursors is also determined by biomechanical cues from the ECM. The differentiation of murine hematopoietic stem cells (HSC) to monocytes/macrophages is mediated through binding to the RGD motifs of fibronectin and other ECM molecules via integrins α5β1 and avb3 [129]. These fate decisions occur within 24 h of ex-vivo culture on ECM-coated surfaces, with greater commitment to a monocyte/macrophage fate demonstrated on stiffer, laminin-coated matrices [129]. This is particularly relevant for the cancer microenvironment, where immature haematopoietic precursor cells arrive at the tumour prior to full differentiation and are therefore highly sensitive to microenvironmental cues such as ECM variation for terminal fate decisions.

Regarding the cancer ECM as a regulator of myeloid cell polarisation, subcutaneous co-injection of cancer cells alongside urinary bladder ECM (UBM) leads to a reduction in tumour growth as compared to control mice co-injected with saline [7]. The survival advantage of UBM-injected mice was attributed to the recruitment of CD4^+^ T-cells and a complex mixture of M1 and M2 macrophages. The majority of TAMs associated with UBM expressed the M2 markers mrc1 and arg1; however, one third of the TAMs were of an M1-polarized, anti-tumour phenotype. Overall, the authors concluded that ECM-mediated TAM polarisation contributed to tumor shrinkage [7]. Related to these findings, cultured macrophages treated with a mixture of solubilized ECM molecules isolated from either porcine small intestine or urinary bladder led to overexpression of M2 macrophage markers CD206, TGM2, fizz-1 and arg1 [130]. This is supported by further studies demonstrating that generation of M2-type macrophages is promoted by the degradation products of ECMs from a range of different organs [8,131,132]. Decellurised dermal matrices for example, promoted the development of Relmα and Arg1-expressing, M2 macrophages characterised by the production of wound healing factors, including MMPs, and growth factors responsible for angiogenesis and ECM remodeling [132].

Some disparity in the effects of specific ECMs on the direction of macrophage polarisation is likely secondary to variation in the experimental settings utilised including the use of 3D vs 2D culture, the precise physical properties and composition of the matrices studied, the source of macrophages and the methods used to define polarisation state [130]. Nonetheless, these findings indicate the sensitivity of macrophages to their physical surroundings in a manner of relevance for cancer ECM biology. Specifically relevant to cancer, culture systems utilising collagen matrices of increasing density promoted the expression of immune-inhibitory chemokines and cytokines by tumour-associated macrophages with resultant inhibition of T-cell mediated cytotoxicity [133]. In murine endometrial cancer, forced expression of the CTHRC1 gene in cancer cells led to the appearance of M2 macrophages within the TME [105] and in the highly aggressive glioblastoma multiforme, expression of the ECM protein osteopontin is associated with adverse outcome and responsible for not only the recruitment of macrophages through a direct chemoattractant effect but also maintenance of macrophages in an M2 pro-tumourigenic state [134].

Neutrophil polarisation is also important for cancer biology with N2 neutrophils providing a pro-tumourigenic phenotype [122]; the cancer ECM may play an important role in modulating neutrophil subtype. Spontaneous breast cancers developed in MMTV-PyVT mice crossed with mice expressing a mutant version of type I collagen that is insensitive to collagenase-based cleavage, demonstrate excessive ECM deposition and accelerated tumour growth compared to those in wild-type mice [135]. Comparison of the immune composite from collagen mutant and wild-type mice failed to demonstrate a difference in absolute cell number, but a significant increase was detected in the proportion of neutrophil-specific cytokines in the collagen mutant murine tumours. Depletion of neutrophils in this model abrogated the tumour-promoting effects of the collagen mutation indicating that collagen in the ECM modulates a pro-tumourigenic phenotype in neutrophils [105,135].

Aside from polarisation, the ECM affects other aspects of neutrophil biology. Neutrophils cultured in 2D on surface-adherent fibronectin, collagen or laminin demonstrated an increase in apoptosis in response to TNFα when compared to neutrophils cultured on PolyHema [136]. This indicates that neutrophils respond to activating cytokines only in the presence of specific ECM proteins and that deposition of specific ECM proteins could be a mechanism through which the cancers inhibit myeloid-based cellular toxicity. This is supported by analysis of human neutrophil cultures which fail to respond to cytokines in suspension but release large quantities of hydrogen peroxide in response to cytokines when cultured on a range of different ECM proteins [137,138]. The significance of these findings, if any, for cancer remains unstudied.

In human breast cancers, myeloid-derived suppressor cells (MDSC) of polymorphonuclear (PMN), or monocytic origin express the ECM glycoprotein SPARC [139]. Loss of SPARC in PMN-MDSCs resulted in reduced tumour growth in two subcutaneous breast cancer models and isolation of splenic PMN-MDSCs from wild-type and SPARC^−/−^ mouse spleens demonstrated that those lacking SPARC were unable to suppress T-cell activation [139]. Contrasting these findings, SPARC expression was down-regulated in malignant melanoma that failed to respond to the chemotherapeutic agent dacarbazine with the authors concluding that expression of SPARC promoted effective tumour immunity [140]. Notably, these findings are only observational and do not describe the mechanisms through which SPARC might alter T-cell mediated tumour immunity or whether SPARC in this setting mediates its effects through mechanisms outside of the immune system such as tumour angiogenesis or through direct effects on tumour cells.

### 5.2. T-Lymphocytes, Natural Killer and Dendritic Cells

T-lymphocytes are the primary cytotoxic effect within the TME and this activity is dependent upon adequate antigen cross-presentation by dendritic cells. Emerging evidence indicates that the ECM regulates this relationship and is therefore directly involved in generation of success or failure of the anti-tumour immune response. The structure and density of the cancer ECM alone can alter the function of infiltrating T-cells [76]. T-cells grown in matrices of high density demonstrate a failure of proliferation and down-regulation of genes involved in cytotoxicity, associated with impaired tumour cell killing [76].

ECM glycoproteins also play a role in modulating T-cell biology in the TME. As discussed above, versican proteolysis is associated with increased T-cell infiltration in colorectal cancer independent of mismatch repair status [108]. One such proteolytic fragment named versikine was able to drive differentiation of CD103^+^ CD11c^hi^MHCII^hi^ dendritic cells expressing high levels of the transcription factors Irf8 and Batf3 [108]. Because these transcriptional programs regulate antigen cross-presentation, this suggests that versikine may promote T-cell cytotoxicity in the TME through effects on dendritic cell phenotype. Support for this hypothesis has been demonstrated in-vivo, where versikine-overexpressing murine lung, breast and myeloma tumours displayed a significant increase in Batf3-expressing dendritic cells when compared to tumours from empty-vector control cells [141]. Analysis of these tumours in the setting of the ovalbumin-OT-I, a model antigen expression system, confirmed an increase in antigen-specific CD8^+^ T-cells in versikine-expressing tumours providing confirmation of the capacity of Batf3-expressing dendritic cells to drive anti-tumour immunity [141].

Osteopontin (SPP1) is a phosphorylated ECM glycoprotein and ligand for integrins that also functions as an immune-modulatory agent in the TME [142]. Expression of SPP1 is associated with poor prognosis in a number of cancers [143]. Importantly, SPP1 is the physiological ligand for CD44, a T-cell surface molecule responsible for regulation of T-cell activation state. Under physiological conditions, the transcription factor IRF8 represses myeloid-derived SPP1 expression; however, in the murine colorectal cancer microenvironment, forced loss of IRF8 led to enhanced production of SPP1 by PMN-MDSCs and tumour cells. SPP1 in turn acted as a T-cell inhibitory ligand through ligation with the CD44 molecule, inhibiting tumour cell cytotoxicity and promoting cancer progression [144].

In murine prostate cancer models, TNC deposited in the ECM inhibited activation, proliferation and cytokine production of infiltrating T-cells [145]. TNC was served as a ligand for α5β1 integrin on T-cells, inhibiting the actin cytoskeleton with a resultant inhibition in T-cell activation. A similar role for TNC has been identified in glioma, where co-culture of glioma cells alongside T-lymphocytes inhibited T-cell activation, again through α5β1 integrin signaling [146]. In humans, TNC was secreted by cancer cells in exosomes, whilst exosomes from glioma patients demonstrated significantly higher TNC expression than those form healthy controls [146].

As well as playing a role in immune cell migration as discussed previously, the galectin protein family is also able to modulate immune cell phenotype. Galectin-3 interacts with the activating receptor NKp30 found on the surface of human Natural Killer (NK) cells [147]. In culture, down-regulation of galectin-3 in cancer cells promoted NK-mediated tumour cell lysis whilst in subcutaneous murine cervical cancer galectin-3 inhibition sensitised cancer cells to death following adoptive transfer of NK cells [147]. Interestingly, galectin-9 knockout mice demonstrate increased IFN-γ production by NK cells and enhanced NK cell degranulation in response to cytomegalovirus infection [148], suggesting that whilst galectin-3 has a negative impact on NK cell function in the TME, the reverse may be the case for galectin-9. In support of this, in murine colorectal cancers, high galectin-9 expression is associated with better prognosis and higher levels of NK cell infiltration [149]. Cultured NK cells demonstrated increased migration in response to galectin-9 through enhanced Rho/ROCK1-mediated F-actin polymerization [149].

## 6. ECM Physics and Tumour Lymphatic Physiology

A further mechanism through which the ECM may affect tumour immunity is by modulation of tumour lymphatics. Lymphatic vessels transport APCs away from tissues and into draining lymph nodes where they can present antigens to T-lymphocytes. In the centre tumour core, lymphatic channels are collapsed, whereas at the interface between the tumour and the surrounding normal tissue, the lymphatics are open and functional [150]. The open lymphatic channels at the edge of the tumour aid lymph node metastasis, whereas the collapsed lymphatics within the TME may prevent transport of APCs and tumour antigens to draining lymph nodes and this may serve to limit effective tumour immunity [151,152].

Collapsed intra-tumoural lymphatics result from increased stiffness and solid stress within the TME; properties determined primarily by the composition and structure of the ECM. The primary physical properties of relevance are stiffness and solid stress. By elevating intratumoural stiffness and solid stress the ECM therefore regulates lymphatic function. This is further compounded by an increase in interstitial fluid pressure within cancers caused by vascular permeability and poor lymphatic drainage [153]. The expected net result of these processes is reduced flow from the tumour lymphatics to the lymph nodes draining the tumour. One would expect this to impair tumour immunity, because APCs carrying tumour-derived antigens are unable to exit the tumour and reach the draining lymph nodes, therefore preventing efficient T-cell priming.

Whilst this process is theoretically sound, it has been difficult to demonstrate experimentally and currently there is little supporting published literature. In melanoma, lymphatic vessels are more abundant at the tumour periphery than within the tumour core and the number of lymphatic channels at the edge of the tumour correlates with the CD8^+^ infiltrate [150], indicating that the presence of lymphatics is important for T-lymphocyte function. In colorectal cancer, the degree of lymphangiogenesis is positively correlated with T-cell cytotoxicity and negatively correlated with metastatic dissemination [154], supporting the findings in melanoma. In a melanoma model, mice lacking tumour lymphatics through loss of VEGFR3, which is responsible for modulating lymphangiogenesis, displayed reduced leukocyte infiltration into the tumour, whilst adoptive transfer of cytotoxic T-cells to these mice inhibited tumour growth [155]. Regarding links to ECM physics, cultured lymphatic endothelia display elevated expression of the transcription factor GATA2 when grown on soft compared with stiff matrices [156]. GATA2 regulates the expression of a range of genes involved in lymphangiogenesis including VEGFR3 [156], indicating that matrix stiffness inhibits lymphangiogenesis by reducing the sensitivity of lymphatic endothelia to its ligand VEGF-C.

### 6.1. Targeting the ECM to Enhance Tumour Immunity

Although therapeutic manipulation of the ECM has been the topic of several recent reviews [157,158], the potential for targeting the matrix to promote tumor immunity is less well recognised. The cancer ECM can be therapeutically targeted in a number of ways including targeting ECM molecules or ECM-remodelling enzymes, altering the structural or physical properties of the matrix, or modulation of fibroblast function as an indirect method to alter ECM deposition.

Identification of the most therapeutically relevant ECM molecules is hampered by ECM complexity both in terms of its structure and molecular diversity, as well as the specific technical challenges that these issues bring with regards ECM isolation and analysis. Equally, given the modular structure of ECM proteins, where certain modules are common across numerous different ECM proteins, the effects of targeting modules on specific proteins may be difficult to predict. Development of methods to successfully extract and purify ECM from tumours [12,159] alongside advances in proteomics and lipidomics have helped to address these difficulties. The fundamental classification of protein subsets within the ECM based on structure and/or function [11] is a further step towards understanding how to therapeutically manipulate the ECM. Below we highlight some of the leading therapeutic candidates for ECM modulation of relevance to cancer immunity.

### 6.2. Renin-Angiotensin Inhibition

Targeting the renin-angiotensin system (RAS) is an established means through which to manipulate the tumour stroma and has demonstrated benefit in various pre-clinical models [160,161]. There are multiple proposed mechanisms through which RAS inhibitors such as losartan and captopril modulate the cancer stroma including inhibition of immunoinhibitory chemokines and cytokines [160], inhibition of aberrant angiogenesis [162] and normalisation of the cancer ECM. Losartan has been shown to inhibit collagen and hyaluronan deposition in various malignancies, resulting in reduced intra-tumoural solid-stress, reduced hypoxia and improved perfusion [157,158]. As indicated by the wealth of data presented in the review thus far, it is likely that these changes will improve tumour immunity through promotion of T-cell migration and inhibition of a myelosuppressive phenotype. Whether this promotes the activity of immunomodulatory therapies remains to be clearly demonstrated, although this is the topic of ongoing clinical trials (Table 1).

### 6.3. Focal-Adhesion Kinase Inhibition

Focal-adhesion kinase (FAK), a tyrosine kinase involved in the regulation of cell adhesion and migration has recently been shown to be an important mediator of the desmoplastic response in pancreatic cancer [163,164]. FAK inhibition using an oral small molecule inhibitor (VS-4718) resulted in a significant reduction in tumour fibrosis associated with a reduction in immunoinhibitory cell populations including granulocytes and myeloid-derived suppressor cells. Crucially, FAK inhibition sensitised pancreatic tumours to immune checkpoint inhibition indicating that modulation of tumour fibrosis has direct implications for tumour immunity [163]. These findings alongside similar reports in other solid malignancies [165,166] have led to the emergence of clinical trials combining FAK inhibitors with immunomodulatory monoclonal antibody therapies, although none have yet reported results and all are early phase (Table 1).

### 6.4. Hyaluronan Depletion

Accumulation of hyaluronan in the cancer ECM promotes solid stress, compresses intra-tumoural blood vessels and reduces perfusion [61]. Administration of a human recombinant hyaluronidase (PEGPH20) reduced hyaluronan accumulation in a murine pancreatic cancer model, improved tumour perfusion and chemotherapy delivery [167]. In clinical trials randomising pancreatic cancer patients to standard chemotherapy with or without PEGPH20, PEGPH20 reduced intra-tumoural hyaluronan levels and improved chemotherapy delivery, but only provided a marginal survival benefit in patients with hyaluronan-rich tumours [168]. However, pancreatic cancer is notoriously resistant to chemotherapy as a result of both stromal and cancer cell-specific resistance mechanisms and so it is reasonable to consider that because hyaluronin depletion improves chemotherapy delivery, it may also improve the success of immunotherapy. This is supported by animal models which demonstrate enhanced delivery of both nanovaccines [169] and therapeutic monoclonal antibodies [170] in the presence of hyaluronidase, whilst clinical trials combining hyaluronidase and immune checkpoint inhibitors are in their early phases (Table 1).

### 6.5. Fibroblast Depletion

Depletion of fibroblasts is an alternative means of affecting the cancer ECM. Fibroblast depletion has primarily focused on the targeting of FAP-positive fibroblast populations and depletion of these fibroblasts reduces tumour ECM in lung and pancreatic cancer models [171]. In the treatment of murine melanoma, ablation of FAP-expressing stromal cells induced a reduction in immuno-suppressive myeloid cells [172], although it is likely that this was related to paracrine effects rather than alteration in the tumour ECM. Unfortunately, a phase II trial of FAP inhibition using the small molecule inhibitor Talabostat failed to demonstrate clinical efficacy in colorectal cancer despite successful enzymatic FAP inhibition [173] and FAP targeting has not been revisited in the clinical literature since.

Importantly, targeting fibroblasts within the tumour microenvironment can also bring about progression of cancer, as has been elegantly demonstrated in the setting of pancreatic adenocarcinoma [174]. Here, depletion of αSMA-positive myofibroblasts unleashed the potential for disease dissemination and metastasis. This indicates that greater understanding of fibroblast heterogeneity both within and between cancer subtypes is required before fibroblast targeting in human tumours can be contemplated. Single cell sequencing is likely to hold the key to this understanding and has already enabled identification of fibroblast subsets linked to aberrant tumour immunity [175,176]. Moving forwards, it will be important to understand whether diversity in fibroblast subset leads to ECM diversity and the implications that such diversity might have for tumour immunity.

## 7. Conclusions

This review has highlighted a significant body of evidence mechanistically linking the tumour ECM to immune cell infiltrate and function. Given the importance of successful immune evasion for cancer progression and metastasis, a greater understanding of how the ECM can be targeted in order to enhance the effect of recently introduced immunotherapies is required. The compositional and structural complexity of the ECM as well as significant intra-tumoural heterogeneity are still yet to be fully understood, although technological advances such as multiplexed-immunohistochemistry, tissue decellularisation techniques and mass-spectrometry are beginning to address these issues. Going forward, it will be important to understand the specific contributions that individual ECM proteins make to matrix function as well as the signaling mechanisms regulating their deposition. Through this process, ECM modulation is likely to emerge as a fundamental cancer therapeutic.

## Figures and Tables

**Figure 1 cancers-12-03331-f001:**
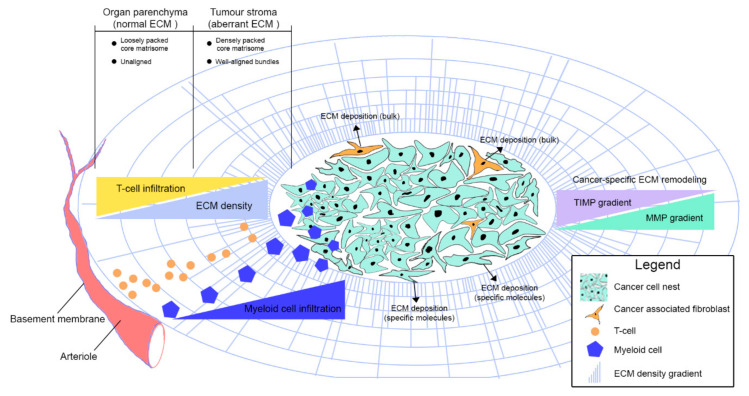
Deregulation of ECM homeostasis in cancer affects immune infiltration. Schematic plan illustrating how growing solid tumours form cancer cell nests to educate tissue-resident fibroblasts that acquire a highly synthetic phenotype leading to the production of densely packed structural ECM components. Cancer cells partially contribute to ECM overproduction by secreting certain laminin chains as well as ECM regulators. The mechanisms underlying ECM remodelling in cancer are complex, with the general line being the subtle balance between ECM-decomposing enzymes (mainly MMPs) and their corresponding inhibitors (mainly TIMPs). One of the major consequences of ECM remodelling in cancer is collagen alignment, which partially regulates immune cell trafficking within the tumour microenvironment. Through this and other mechanisms, the cancer ECM excludes some immune cell subsets (such as infiltrating CD8^+^ T cells) whilst enabling active infiltration of others, such as macrophages and neutrophils. ECM extracellular matrix.

**Figure 2 cancers-12-03331-f002:**
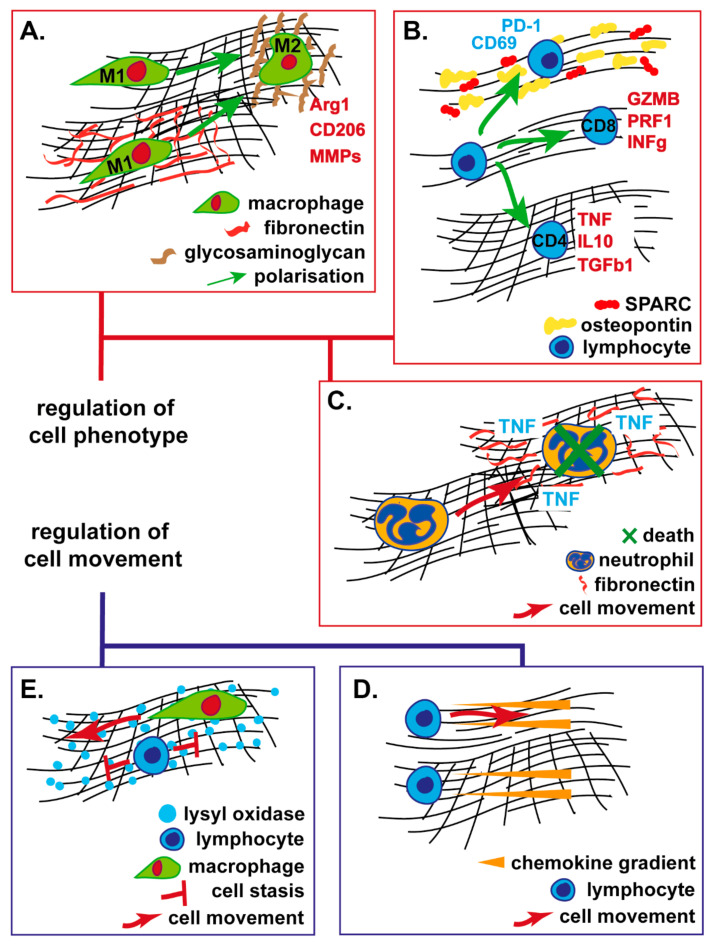
Proposed mechanisms though which the tumour ECM modulates immune cell phenotype and movement (see text for further details). (**A**) Macrophages in dense matrices demonstrate alternative states of activation dependent on the presence of various ECM glycoproteins. (**B**) T-lymphocyte phenotype is dependent upon collagen fibre density with loose matrices supporting cytotoxic T-lymphocytes but dense matrices or those incorporating specific glycoproteins leading to immune-inhibitory phenotypes. (**C**) Neutrophil survival and cytokine production is dependent upon the presence of specific ECM glycoproteins. (**D**) T-lymphocyte movement is driven by chemokine gradients in loose ECM, but in dense ECMs, T-lymphocytes to not demonstrate chemokine-directed movement. (**E**) Cancer ECM structure may differentially modulate the migration of T-lymphocytes and macrophages, with only macrophages able to move in dense, highly cross-linked matrices. Collagen fibre matrices demonstrated in black with degrees of density or cross-linking indicated. Arg Arginase, MMP matrix metalloproteinase, GZM granzyme, PRF perforin, INF interferon, TNF tumour necrosis factor, IL interleukin, TGF transforming growth factor, SPARC Secreted Protein Acidic and Rich in Cysteine.

**Table 1 cancers-12-03331-t001:** Examples of clinical trials combining agents targeting the tumour ECM with immune checkpoint inhibition. FAK focal adhesion kinase, RAS renin-angiotensin system.

Target	Cancer	Drug	Additional Treatments	Setting	Trial Number	Phase	Primary Outcome/Aim
FAK	Pancreas	Defactinib	PD-1 (Pembrolizumab)	Neoadjuvant and adjuvant	NCT03727880	2	Pathological response
	Lung, mesothelioma, pancreas	Defactinib	PD-1 (Pembrolizumab)	Palliative	NCT02758587	1 and 2	safety
	Advanced solid cancer	Defactinib	PD-1 (Pembrolizumab), Gemcitabine	Palliative	NCT02546531	1	Dose escalation
RAS	Pancreas	losartan	PD-1 (Nivolumab), FOLFIRINOX, SBRT	Neoadjuvant	NCT03563248	2	R0 resection rate
Hyaluronin	Stomach, lung	PEGPH20 (PEGylated recombinant human hyaluronidase	PD-1 (Pembrolizumab)	Palliative	NCT02563548	1	Safety, dose escalation
	Metastatic pancreas	PEGPH20	PD-1 (Pembrolizumab)	Palliative	NCT03634332	1	Progression-free survival

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
