# Peer review of "Cancer Extracellular Matrix Proteins Regulate Tumour Immunity"

_cancers, 2020, doi:10.3390/cancers12113331_

Round 1
Reviewer 1 Report
"The Cancer Extracellular Matrix Regulates Tumour Immunity" is a well written and light review / introduction to the topic. I think it is a respectable contribution. There are two areas of improvement that I think are highly relevant that I would like to point out to the authors:
DDR - collagen - prolifieration
fibrillar collagen - immune interactions
There is a little bit of coverage of the latter and no mention of the former although there is discusion of integrin, mmp, timp exchanges. There really should be some discourse, paragraph at the bare minimum on DDR.
https://www.mdpi.com/1422-0067/21/11/4166
https://www.sciencedirect.com/science/article/abs/pii/S0167488919300795
The above are high level coverage of both of these topics and I think they compliment and augment this manuscript, being directly relevant to proliferation and cellular immunity.
Author Response
Reviewer 1
- "The Cancer Extracellular Matrix Regulates Tumour Immunity" is a well written and light review / introduction to the topic. I think it is a respectable contribution. There are two areas of improvement that I think are highly relevant that I would like to point out to the authors:
DDR - collagen - prolifieration
fibrillar collagen - immune interactions
There is a little bit of coverage of the latter and no mention of the former although there is discusion of integrin, mmp, timp exchanges. There really should be some discourse, paragraph at the bare minimum on DDR.
https://www.mdpi.com/1422-0067/21/11/4166
https://www.sciencedirect.com/science/article/abs/pii/S0167488919300795
The above are high level coverage of both of these topics and I think they compliment and augment this manuscript, being directly relevant to proliferation and cellular immunity.
Response.
We thank the reviewer for positive evaluation of our work. We have carefully reviewed the articles provided above and have incorporated them into our manuscript following their discussion.
Reviewer 2 Report
The authors focus on the role of extracellular matrix (ECM) in tumor immunobiology. The subject is timely and important as the effects of ECM need to be taken into account when developing novel therapeutic strategies. The text has to some extent responded to the intended scope. Prior to publication, however, of certain aspects need to be clarified and role of other ECM components discussed.
Comments:
- Line 71. The authors need to clarify the term -stroma. Stroma is not the interstitial matrix but matrix together with the embedded connective tissue cells, blood vessels, and inflammatory cells. Please correct throughout.
- Line 71. It would be more correct to define matrix compartments as pericellular, territorial (extracellular) and basement membranes as each respective compartment has a discrete composition and role both in homeostasis and cancer progression.
- Proteoglycans are an important component of the cancer ECM and need to be separately discussed. E.g syndecans affect both breast cancer matrix density and immunobiology. Importantly, thus, the cancer stroma density does not depend just on fibrillar protein content but also on proteoglycan expression. please discuss.
- Many cancer types are hormone-dependent, which is immediately correlated to their ECM composition and tumor immunity. This needs to be, at least, mentioned.
- The therapeutic aspects of ECM modification need to be discussed.
- The title of the review -The Cancer Extracellular Matrix Regulates Tumour Immunity-, is not representative of the text. The authors discuss the role of ECM proteins, their expression and remodelling on various aspects of tummor immunity. The very important effects of hyaluronan on tumor immunobiology are not addressed. Since including HA would probably be overambitious the authors might consider changing the title of the manuscript. e.g The Cancer Extracellular Matrix Proteins Regulate Tumour Immunity
Author Response
- Line 71. The authors need to clarify the term -stroma. Stroma is not the interstitial matrix but matrix together with the embedded connective tissue cells, blood vessels, and inflammatory cells. Please correct throughout.
Response.
We completely agree with this point and have removed use of the word stroma in line 71.
- Line 71. It would be more correct to define matrix compartments as pericellular, territorial (extracellular) and basement membranes as each respective compartment has a discrete composition and role both in homeostasis and cancer progression.
Response.
We have highlighted the organisation definition of the matrices incorporating the words pericellular, basement membrane and interstitial matrix is helpful because breaching of the basement membrane helps define a malignant from a benign tumour. The disorganisation of the matrix in cancer has primarily focused on changes in the basement membrane as an initiating event and also alterations in the inter-territorial matrix.
We have included a new section on the pei-cellular matrix but have maintained the descriptions of the basement membrane. We have changed the description of the ‘stromal’ ECM to the inter-territorial ECM.
- Proteoglycans are an important component of the cancer ECM and need to be separately discussed. E.g syndecans affect both breast cancer matrix density and immunobiology. Importantly, thus, the cancer stroma density does not depend just on fibrillar protein content but also on proteoglycan expression. please discuss.
Response.
We have discussed the role that a number of ECM glycoproteins play in modulating the immune infiltrate in cancers.
Please see section entitled ‘Non-structural ECM proteins’ which focuses predominantly on glycoproteins and includes description of the following glycoproteins: HAPLN1, Tenascin C, Thrombospondin 1, CTHRC1, fibronectin, osteonectin, periostin and versican.
We have also included discussion of the specific interactions between immune cells and particular glycoproteins in the sections entitled ‘ECM Regulation of Immune Cell Function’.
Some of the interactions between glycoproteins and immune cells have been highlighted in Figure 2.
We do not feel it is possible to provide a complete list of all of the studies demonstrating links between glycoproteins and immune cells and so have highlighted those where we feel the most compelling evidence exists.
- Many cancer types are hormone-dependent, which is immediately correlated to their ECM composition and tumor immunity. This needs to be, at least, mentioned.
Response: There are a large number of cancer types that are not hormone dependent (pancreatic cancer, colorectal cancer etc) and these also demonstrate stromal activation. The authors therefore do not feel that it is necessary to specifically highlight the relationship between hormone stimulation and ECM, particularly as they have now made significant additions to the manuscript.
- The therapeutic aspects of ECM modification need to be discussed.
Response: Therapeutic targeting of the cancer ECM has been covered in numerous recent reviews and we have now referenced some of these to draw them to the readers attention.
We have also included a section highlighting the strategies through which targeting the ECM may diver the greatest immunological benefit focusing specifically on the targeting of FAK, hyaluronan, the renin-angiotensin system and specific fibroblast populations.
- The title of the review -The Cancer Extracellular Matrix Regulates Tumour Immunity-, is not representative of the text. The authors discuss the role of ECM proteins, their expression and remodelling on various aspects of tummor immunity. The very important effects of hyaluronan on tumor immunobiology are not addressed. Since including HA would probably be overambitious the authors might consider changing the title of the manuscript. e.g The Cancer Extracellular Matrix Proteins Regulate Tumour Immunity
Response: We have changed the title of the manuscript to: Cancer ECM Proteins Regulate Tumour Immunity.
Round 2
Reviewer 1 Report
I am content to support the publication of this reasonable review. It is a work horse contribution and is not remarkable, but hardly not worthwhile either.
I noted some spelling / grammatical issues with the newly added text sections, missing words etc.
Further, although I had asked for specific topics to be addressed, the authors have done so with a sentence or so and the references I suggested... My intention had been to encourage more disscusion of those areas to elevate and improve the review as a whole, those papers were starting points to find more to include in their own work.
If there is room in the authors workflow to do this, I would encourage them to do so. However, at this point I can sign off with an 'okay'.
Best regards
Author Response
Many thanks for the further comments.
We have now added additional information RE DDRs and immune cell motility as per your request (see lines 213 onwards).
We have gone over the added text to provide grammatical/spelling corrections.
Reviewer 2 Report
The authors have addressed the raised concerns.
Author Response
Nothing further to add.